# Precision Beekeeping Systems: State of the Art, Pros and Cons, and Their Application as Tools for Advancing the Beekeeping Sector

**DOI:** 10.3390/ani14010070

**Published:** 2023-12-24

**Authors:** Pier Paolo Danieli, Nicola Francesco Addeo, Filippo Lazzari, Federico Manganello, Fulvia Bovera

**Affiliations:** 1Department of Agriculture and Forest Sciences (DAFNE), University of Tuscia, Via S. C. de Lellis snc, 01100 Viterbo, Italy; f.lazzari92@gmail.com (F.L.); federico.manganello@unitus.it (F.M.); 2Department of Veterinary Medicine and Animal Production, University of Napoli Federico II, Via F. Delpino, 1, 80137 Napoli, Italy; bovera@unina.it

**Keywords:** honeybee, sensors and systems, beekeepers, internet of things, literature review

## Abstract

**Simple Summary:**

This review aims to raise attention to precision technologies applied to the world of beekeeping, the implementation of the precision technological approach by a normal beekeeper, the reliability of the data analysis, the state of the art, and the pros and cons found to date in the apiary sector.

**Abstract:**

The present review aims to summarize the more recent scientific literature and updated state of the art on the research effort spent in adapting hardware–software tools to understand the true needs of honeybee colonies as a prerequisite for any sustainable management practice. A SWOT (Strengths, Weaknesses, Opportunities, and Threats) analysis was also performed with the aim of identifying the key factors that could support or impair the diffusion of precision beekeeping (PB) systems. Honeybee husbandry, or beekeeping, is starting to approach precision livestock farming (PLF), as has already happened in other animal husbandry sectors. A transition from the current paradigm of rational beekeeping to that of precision beekeeping (PB) is thus expected. However, due to the peculiarities of this species and the related farming practices, the PB technological systems (PB systems) are still undergoing a development process that, to some extent, limits their large-scale practical application. Several physical–chemical (weight, temperature, humidity, sound, gases) and behavioral traits (flight activity, swarming) of the hive are reviewed in light of the evolution of sensors, communication systems, and data management approaches. These advanced sensors are equipped with a microprocessor that records data and sends it to a remote server for processing. In this way, through a Wireless Sensor Network (WSN) system, the beekeeper, using specific applications on a personal computer, tablet, or smartphone, can have all the above-mentioned parameters under remote control. In general, weight, temperature, and humidity are the main hive traits monitored by commercial sensors. Surprisingly, flight activity sensors are rarely available as an option in modular PB systems marketed via the web. The SWOT analysis highlights that PB systems have promising strength points and represent great opportunities for the development of beekeeping; however, they have some weaknesses, represented especially by the high purchasing costs and the low preparedness of the addressed operators, and imply some possible threats for beekeeping in terms of unrealistic perception of the apiary status if they applied to some hives only and a possible adverse impact on the honeybees’ colony itself. Even if more research is expected to take place in the next few years, indubitably, the success of commercial PB systems will be measured in terms of return on investment, conditioned especially by the benefits (higher yields, better colonies’ health) that the beekeeper will appraise as a consequence of their use.

## 1. Introduction

One of the most important challenges of this and, probably, the next century is to safeguard biodiversity around the world. In this regard, great attention has been focused on insects, whose decline on the local and global scales has been reported [1]. Insects are responsible for a wide range of functional roles [2,3,4,5], one of which is pollination; in fact, the pollinators can improve the production of 70% of the globally most important crop species and influence 35% of the global human food supply [6]. Medium- and long-term strategies can be studied to control the decline of wild pollinators [7], but, in the short-term, one of the possible strategies is to increase the number of farmed pollinators, in particular honeybees (*Apis mellifera* L.). In this context, honeybees and beekeepers are two important resources for the Earth: insects for their pivotal role in pollination, and humans for their ability to protect and preserve the health and survival of the honeybees. However, the famed honeybee is also exposed to several risks, such as the use of pesticides for intensive crops [8,9,10,11], climate change, and the impact of pathogens (parasites, bacteria, and viruses) responsible for a wide range of illnesses. Beekeepers try to contrast these risks by using appropriate farming techniques, different kinds of chemicals, and/or drugs; however, the results are not always satisfactory [12,13], and the health and production of the honeybee are often negatively affected. To obtain better production (and better insect health) from a hive, it is important to have proper management, as it is well known that regular inspections of the colonies can improve honey production and thus the remuneration of the farming activity. However, it is important to underline that, in some circumstances, manual visits to the hive can have negative impacts on the colony [14]. In addition, very often the beekeepers could not check their hives with regularity, for example, due to climate conditions or low time availability. In this context, modern technology can be a valuable help for beekeepers. The term “Precision Beekeeping” (PB) has been defined for the first time by Zacepins et al. [15] as a strategy for the management of an apiary based on monitoring individual honeybee colonies to minimize resource consumption and maximize productivity. This new approach can be considered a new evolutionary phase of beekeeping, as indicated by Zogovic et al. [16]: from “traditional beekeeping”, through “rational beekeeping”, up to “Precision Beekeeping”. The PB is organized into three key points: data collection, data processing, and the data output phase [17]. As recently reported by Alleri et al. [18], data can be captured by different kinds of sensors for monitoring, for example, weight, temperature, humidity, sounds, etc. The sensors can be placed inside or outside the hive; in this way, the hive becomes “smart”. In this way, the data is checked in real-time through specific applications on different devices (personal computers, tablets, and smartphones) and processed thanks to algorithms. Beekeepers can thus consult and download the data obtained from their archives. The data processing is important to define if a condition measured at a specific time falls or is not within a “normal” range. In addition, the processing data system can produce signal alarms if some parameter is out of the “normal” standard. From a practical perspective, PB is not only the application of intelligent technologies to hive bee farming but also an in-depth knowledge of the various genetical, nutritional, physiological, and physio-pathological aspects linked to the world of bees. Precision beekeeping is at the beekeeper’s service, but the beekeeper must be able to properly read the indications that the digital system provides him, including the alarm signals. Technology is a very important aid, but it is not infallible; therefore, it can support the beekeeper but not replace him, because obviously, it is up to him to make the final decision on any corrective measures to be taken in the presence of anomalous situations. All in all, it seems clear that talking about a multifaceted topic such as precision beekeeping is not easy. The purpose of this review is to illustrate the most recent developments in technologies applied to bee production and how they can be linked to the genetic, physiological, ethological, nutritional, and pathological aspects to arrive at rational management of the apiary for the protection of animals and profitability of productions. A SWOT (Strengths, Weaknesses, Opportunities, and Threats) analysis was also performed with the aim of identifying the key factors that could support or impair the diffusion of precision beekeeping systems.

## 2. Data Collection and Storage: Sensors and Systems

In PB, data collection is the first and probably one of the most important steps. It is typically performed using sensors integrated into beehives and connected to the main processing system [19]. A sensor (or node) is a device sensitive to a specific physical–chemical cue and, if solicited, can generate an electrical signal [20]. These advanced sensors are equipped with a microprocessor powered by batteries; the microprocessor records the data and, thanks to a Wireless Sensor Network (WSN) system [21], sends it to a remote server for processing [18]. Sensors, microprocessor, battery, and remote server are part of the “system” that is completed by the device of the beekeeper (personal computer, smartphone, or tablet) in which a specific application shows the data to the user on demand and sends signals of alarm if a parameter, for example, the temperature, the weight, the relative humidity, the buzz sounds, the gases, as well as the honeybee activity and the location of the hive, are outside the normal range [18,22]. Thus, the type of sensor (and thus its accuracy) is very important. It is possible to use very simple and low-cost systems. The problem is whether these systems can achieve useful control over the hive. For example, Romanov [23] used an on-site bee colony approach to evaluate the hive temperature, in which a small digital sensor is placed inside the hive and connected to an external display where the temperature is reported. This very simple system is not able to send the data via the web, and the temperature can be observed remotely only by placing a camera near the digital display. This system may seem functional, but the main problem is that the data cannot be stored [19], and it is possible to only do on-site or on-time observation without an alarm signal available for beekeepers. The progress of these technologies allows for different and more flexible systems to record, store, and process data. For example, the data recorded by sensors can be transmitted by wired or wireless connection to a PC on the apiary [15,24]. This kind of solution allows for more information to be given to the beekeeper, but to record and send the data, the PC must be turned on (energy expenditure). Another possibility is double-data sending. The sensors recorded the data and transmitted them to the apiary PC, which is connected to a remote service or a cloud device [25]. In this case, the local PC is only an intermediary, and the data storage and processing are from the remote service. The remote service will send the data to the beekeeping devices. When the apiary’s PC is turned off, it is possible to store the data on the remote server, but they cannot be processed. Another approach relies on micro-controller platforms to collect and send the data to the remote controller. The micro-controller platforms need less energy and thus are more stable than a PC, but if the system is turned off, the data processing is interrupted, and no data or alarms are sent to the beekeeper’s devices. However, more modern technologies are producing sensors able to connect to the web using 4G or 5G connections. In this way, recorded data is sent directly to the remote server for storage and processing, and only the interruption of the network or the breakage of the sensors can interfere with the monitoring of the hives. All this is possible where the apiary is reached by a 4G or 5G connection (even 3G in some sensors). Though networks are by now fairly spread in rural areas, it is possible that some areas are not completely covered by the service. In addition, with this kind of approach, each sensor point needs a battery as a source of energy and a SIM for the connection to the network. It means that each hive beekeeper must provide these resources.

### 2.1. Weight

The weight of a beehive fluctuates according to the season (the lowest during winter, the highest during the productive period) and can be a valuable predictor of honey production and, in general, of the activity of the hive. The weight is probably the easiest parameter to measure by using, in general, non-invasive sensors. The different systems available on the market use a scale consisting of load cells, which rely on mechanics and resistive theory, integrated with amplifier modules that are linked to the microcontroller [26]. The load cells are made of a material that moves back when pressure (weight) is applied. One used technology is a “resistive strain gauge” in the load cell, which makes it possible to measure the resistance (Ohm) of the cell [27]. An important aspect when working with load cells is the temperature and humidity around the cell, which will make the value change for the same load, resulting in different measurement values [27]. The load cells can be converted into dedicated electronic circuits (Analog to Digital Conversion, ADC). The ADC circuits are standardized, and one interesting characteristic of ADC converters is the resolution, defining how detailed values the ADC will provide as output. For accurate, reliable measurements and ease of use, the weight sensor or weighing scale is placed under a hive, where each hive can comprise multiple supers/chambers. A single honey chamber can weigh up to 30 kg, which means that a hive with three honey chambers and a brood chamber can weigh up to 120 kg. The weighing scale needs to have an appropriate range of measurements and must be sensitive enough to detect daily changes in the weight of the hive with a resolution of a few grams (in general, less than 100 g) [28]. The choice of hive type also plays a role in this variation. For instance, beekeepers may opt for Dadant hives, which are characterized by their built-in base, greater capacity, and, albeit heavier weight. Alternatively, they may choose Langstroth hives, which offer reduced weight and enhanced versatility but necessitate more meticulous oversight from the beekeeper [29,30]. The weighing scale is often the only major component sitting outside the hive. It can be configured to operate independently by having its own power source, microcontroller, and communication system, or it can be configured to draw power from the monitoring system within the hive and relay detector data back to the monitoring system via wired connections. Drawing power from the internal system and communicating data using wired connections is cost-effective, but the external wiring of such a setup hinders hive inspections and transportation. On the other hand, a stand-alone system provides a lot of ease of use but equipping each hive with such a scale can be prohibitively expensive. As a special case, Sakanovic and Kevric [31] proposed an innovative system able to record the weight of each single frame inside the hive: a connected frame holder inserted between the roof and the base of a beehive at the top of the frames. The authors stated that the system is not so invasive and does not affect the behavior of the honeybees. That could have very interesting applications not only from a scientific point of view but also for the beekeepers because it would be possible to know the increase in brood or forages.

### 2.2. Temperature

Temperature was one of the first parameters recorded in a hive. Almost a century ago, in 1926, Dunham [32] conducted one of the first experiments by using eight thermocouples placed in different sites of the hive and manually recording the temperature each hour [33]. The temperature is a very important parameter for bee colonies, as its detection can be used to identify different conditions in the hive, such as brood development, the pre-swarming condition, and, in general, the health status of the hive. The temperature of the hive can be monitored by using different systems. Cook et al. [34] equipped the experimental hives with four temperature sensors located in the middle of the frames, from the central one to the outskirts of the hives. In hives with active bee colonies, there was an average gradient of 0.03 °C/cm in the observed temperature range (*p* < 0.001). In hives with no bees, this slope was reduced from 0.025 °C/cm (*p* < 0.001) to 0.005 °C/cm [34]. However, based on a total of 776,872 observations, the authors concluded that the variation in the time to reach the peak temperature at different sensor points was not significant [34]. In general, the most common choice is to place a single temperature sensor in the middle of the hive, where the queen bee is [35]. There are several concerns about the possibility that the physical obstruction of the sensors can affect the activities of the honeybee inside the hive. For this reason, Meikle et al. [36] placed the sensors in the upper part of the hive, and Giammarini et al. [37] chose the inner side of the beehive base rather than directly on the frames. In both cases, the accuracy of the data might be lower than the on-frame alternative [22].

### 2.3. Sounds

Sounds, produced through body vibrations, are one of the methods used by honeybees for on-colony communication [38]. The detection of specific sound signals produced by honeybees can be useful in assessing physiological or pathological conditions in the hive. The honeybees can produce sound with frequencies ranging from about 0 up to a few thousand hertz [22]. Thus, the first requirement for an acoustic sensor to be included in the hive is to work in this frequency range. Another very important point is to choose the most appropriate acoustic sensor, as the technology supplies different kinds of microphones: electret microphones, a type of electrostatic capacitor-based microphone that eliminates the need for a polarizing power supply by using a permanently charged material, used by Qandour et al. [39] and Anand et al. [40]; and microelectromechanical system (MEMS) microphone, a micro-scale device used, for example, for smartphones, that provide high-fidelity acoustic sensing and is small enough to be included in a tightly integrated electronic product [41,42]. In the past, another system has been studied to measure the vibration in the hive: the laser Doppler vibrometer proposed by Michelsen et al. [38]. The laser Doppler vibrometer (LDV) is a tool that measures such vibrations by directing a laser at a surface and comparing the frequency of the returning light to an internal reference beam [43]. This system is very precise and has several applications in different kinds of industries, but it is also too expensive. In fact, no other research is available in the literature on the application of this system for vibration detection in the hive. As for the sensor for temperature, microphones must be placed inside the hive, but in a different position. In the system described by Cecchi et al. [44], the microphone is located in a protective box on the backside of a hive. Robles-Guerrero et al. [45] placed the microphones on the roof of the hive, on the top of the frames. Ferrari et al. [41] sited the microphones inside the hives together with a temperature and humidity sensor to obtain an early prediction of swarming. All the researchers using microphones for sound detection considered it useful to protect the sensors from propolisation through nets, plastic, or metal boxes [39,40,41,42,44,45]. Recently, another possibility to evaluate the sounds in the hive has been proposed by using accelerometers, devices that measure the vibration, or acceleration of motion, of a structure. Bencsik et al. [46,47] and Ramsey et al. [48] used accelerometers to evaluate the swarming and the queen activity, respectively. The use of accelerometers has several advantages compared to microphones: accelerometers can stay in a hive for several years, and there are very few problems tied to propolisation; it is possible to detect vibrations with a high precision signal at a low and very low frequency; all considered, probably monitoring vibrations makes more sense than the detection of sounds [49]. Ramsey et al. [48] used two accelerometers placed directly on the frame after creating a cavity at the center of the frame. Those authors indicated that after two seasons of use, no negative effects were detected on the colonies. However, it is important to prevent contact between the bees and the metal components. Aumann et al. [50] stated that vibration sensors, when mounted on the outside wall of a beehive, are capable of detecting swarming and robbing activity. However, as recently indicated by Uthoff et al. [51], more work needs to be undertaken to develop a robust classification of sounds and vibrations in the hive, at least for the detection of the queen or of the swarming activity. As an example, during parasitic or other infections in the hive, as well as during warning situations, sounds with a range frequency between 300 and 3600 Hz are produced [52]. At the current state of the research, it is not possible to obtain accurate information on the different behaviors with vibration detection alone.

### 2.4. Images

The collection of images is an interesting and very useful approach. By acquiring videos of pictures, the beekeeper can directly see the honeybees and, in some cases, detect a problem without the necessity of data processing. Optical cameras have, in general, been placed at the entrance of the hive to monitor the honeybee “traffic”, including foraging, surveillance, and fan activities. Edwards-Murphy et al. [27] tried to use an infrared camera inside the hive to monitor the activity of the family, but this system had no further applications, probably due to technical problems. To control the honeybee traffic, the external camera must be placed at an adequate distance from the hive entrance. Crawford et al. [53] placed the cameras 27.4 cm from the hive entrance; Yang and Collins [54] placed the camera 30 cm above the hive entrance; and, to permit better vision of the animals, the platform was painted green; and Sledevic [55] placed the cameras 40 cm above the hive. Kulyukin and Mukherjee [42] placed the cameras on top of supers (one or two with a distance camera-landing platform of 35 and 60 cm, respectively) and showed no problems with computer vision and the consequent algorithm application. Shimasaki et al. [56,57] placed the cameras in front of the beehives. All those approaches have proven to be effective for monitoring honeybee activity, but their efficacy depends on the resolution of the camera used. As a specific type of camera, thermal cameras can also be used to monitor the temperatures of the honeybees or the hives, with the aim of detecting temperature changes in pre-swarming conditions [58,59]. To date, no thermal camera has been used to count the incoming and outgoing activity at a hive entrance [60] due to its high cost, low resolution, and low frame rate in comparison to optical cameras. Probably, this kind of system can have future development as the technology recently produced low-cost thermal cameras for medical applications [61]. In fact, Williams et al. [60] stated that thermal cameras are a contender for bee counting applications and can be better than optical cameras in poorly lit conditions and without visual aids. Images can also be used for the early detection of honeybee pathology through a visual examination of the animals. Under laboratory conditions, Elizondo et al. [62] used thermal cameras to study the movement of the mite in the brood. Schurischuster et al. [63] used a visual system to count the mites on adult honeybees. For this purpose, honeybees were forced to enter the hive through narrow tunnels, and artificial light was used to create better conditions for image processing. Braga et al. [64] proposed an interesting grading scale to evaluate the health status of adult honeybees from the morphological aspect of the animals, obtaining 95% accuracy for the health classification of a bee and 82% accuracy in detecting the presence of bees in an image.

### 2.5. Gases

The air inside the hive is a complex mixture of many different volatile compounds released by the honeybees (e.g., pheromones, other chemicals released to repel pests and predators, metabolites, etc.), within the hive (from honey, nectar, larvae, beeswax, pollen, and propolis, or materials out of which hives are constructed), and external sources (from vehicles, farms, industries, and households in the vicinity of hives) [65]. Each hive has an individual gas profile. Considering that the other gas producers are, in general, constant, changes in gas composition inside the hive are tied to changes in gas emissions by honeybee adults or larvae. The simplest gas to detect is carbon dioxide, whose increasing concentrations can indicate, for example, an increase in workers’ population in the hive and, over a certain limit, can indicate an inappropriate environment for honeybees [66]. Metal oxide semiconductors (MOx) are gas sensors whose small size allows their placement in the hive [67]. One family of these MOx sensors responds to give an estimate for CO_2_. These sensors have a low power consumption [68], but the cost is still high, so a practical application in beekeeping is difficult at this stage. In addition, MOx sensors can detect a wide range of gases, so a specific calibration is mandatory to understand what kind of gas the sensor is measuring at a specific time [69,70]. Another solution for CO_2_ detection in the hive is the use of nondispersive infrared (NDIR) sensors. The basic principle is that an infrared (IR) source, closely matched to the absorption frequency of CO_2_, shines down a sample tube containing air. The difference between the amount of light radiated by the IR source and the amount of IR detected is directly proportional to the number of CO_2_ molecules in the air sample in the tube [70,71]. Unlike MOx sensors, this is highly selective for CO_2_ and can be calibrated to directly measure the concentration of CO_2_ in ppm [67]. However, to date, the availability of this kind of sensor is limited because the industry has focused its production on MOx sensors.

### 2.6. Humidity

The measurement of humidity is readily available in low-cost, small capacitive sensors that provide a high level of accuracy both in analog and digital formats [72]. Digital devices usually include temperature measurements in the same package and reduce measurement errors by undertaking Analog to Digital Conversion on the sensor chip rather than introducing possible noise in the measurements [71,73]. Easily deployable breakout boards start for as little as EUR 8, with the raw chips costing even less. Many manufacturers produce these with inter–integrated circuit (i2c) interfaces, each with different addresses, allowing some spatial variation over a hive to be monitored with relatively simple and inexpensive hardware [67].

## 3. Internet of Bees and Data Management

In an article published in the online journal *Make Magazine*, the term “Internet of Bees” (IoB) is reported to indicate the application of IoT to beekeeping. The data collected with the different kinds of sensors applied to the hive must be stored in a system (from a local PC to a cloud). As more data are collected, the more precise will be any sort of prediction of hive behavior or activity. From this point of view, a great role plays in the possibility of creating archives of big data and the possibility of sharing the data collected from different apiaries. Different datasets can be available in the libraries of the companies producing PB systems, where data is protected from other web users. The working process of data can be described by using the OODA loop as described by Brehmer [74] and Atwood [75]: Observe, Orient, Decide, Act. Also, in traditional and rational beekeeping, the same loop is used, but with manual or semi/automated inspection instead of using electronics, like IoB devices. All IoB devices need internet connectivity. There are multiple techniques used, including Wi-Fi, Bluetooth, Mobile Internet (5G, 4G, or 3G), Fibre Broadband, etc. The development of embedded electronics like Arduino^®^ and Raspberry Pi^TM^ has created new opportunities to have a low-cost, standardized device to use as an IoT device. The processing phase of bee colony data is typically limited to basic statistical analysis [76] to determine such bee colony states as queenlessness, bloodlessness, pre-swarming, swarming, and after-swarming. The data output phase includes methods to provide processed data—information—to the end user in the form of a graphical or tabular representation. As recently underlined by [77], the use of wireless network technologies in PB has some limitations, represented by data imperfections, granularity, or inconsistency, but also technical problems such as internet or mobile network coverage. To reduce mistakes that can generate false alarms, it is possible to use multi-sensors with additional data sources. However, this solution poses another challenge: processing data from different sources as a singular unit. This kind of advanced approach is defined as data fusion. Data fusion is the process of integrating multiple data sources to produce more consistent, accurate, and useful information than that provided by any individual data source. In the OODA approach, orienting and deciding acts are possible with the application of specific mathematical models (algorithms) able to analyze data and predict possible evolutions or consequences of a specific situation. Machine Learning (ML) integrates statistics and computer science to build algorithms that are more efficient when they are subject to relevant data rather than being given specific instructions [78]. Different mathematical models can be used to process the data, and the precision of the output depends on the quantity and quality of the data as well as the accuracy of the prediction model. Brini et al. [79] used different tree methods: Random Forest [80], Extreme Gradient Boosting (XGB) [81], and a regression tree [82] to analyze data on hive weight and concluded that the first two methods outperformed linear models when predicting the hive weight variation. Andrijevi et al. [83] applied mathematical models based on recurrent neural networks to the data obtained from a bee counter at the hive entrance and showed a very high accuracy of the prediction. Pham et al. [84] tested different algorithms to process data on the foraging behavior of honeybees and observed that all were highly competitive in terms of learning accuracy and speed. Dimitrios et al. [85] tested three different classification algorithms: the k-Nearest Neighbors algorithm (k-NN) and Support Vector Machine (SVM), and a newly proposed by the authors, U-Net Convolutional Neural Network (CNN), developed for biomedical image segmentation. The results show that k-NN and SVM, which are already used for bee sound analysis processes, provide the most accurate results for late and early detection of swarming, respectively. Focusing on early detection, which can alleviate or prevent the event, our experiment showed that SVM is the most appropriate method, while k-NN fails to detect it accurately. For early detection of swarming events, U-Net CNN performs almost as well as SVM and has the potential to perform even better with frequency-targeted data input and model parameter fine-tuning. The authors set, as future work, the extensive evaluation of the proposed U-Net CNN algorithm fine-tuning towards swarming events and the extension of their experiments to other deep learning algorithms.

## 4. Study Cases and Market Options

In the last decade, PB system research and development transformed beehive management, enhancing bee health and apicultural efficiency. These breakthroughs have laid the groundwork for the contemporary commercial products we have today. Smart PB systems that meet the needs of beekeepers to remotely check their hives/apiaries have been developed within the paradigm of PB. Some case studies can provide an overview of real-world examples that highlight the practical applications of cutting-edge technologies and beekeeping practices.

### 4.1. Study Cases

Monitoring of internal hive temperature and humidity was the focus of the investigation conducted by Rodriguez et al. [86], who constructed their PB system, known as myBee, based on the system architecture outlined by Kviesis and Zacepins [19]. This system underwent testing on ten beehives within a farm environment, and the internal temperature data collected within the hives were validated using an infrared camera system. The study was aimed at showing that the use of a PB system may have many advantages, such as the need not to handle each beehive to identify the colony’s conditions and to process the data to obtain reports and statistics, thus detecting any unfavorable conditions in the beehive in real-time. Shaw et al. [87] monitored the internal temperature of bee hives by using a thermal camera mounted in front of the hive. The images had a radiometric uncertainty of 0.5 W m^−2^ sr^−1^ when the camera was allowed to stabilize for at least 20 min before recording. In those experimental conditions, the radiance of images was highly correlated (r^2^ = 0.62) with the normalized manual counting (e.g., storage frames, brood frames). As a result, by applying a simple data correction model, it is possible to accurately estimate the demography of a colony without making manual inspections, which turns out to be very promising in the winter when it is not always possible to visit the colonies. To better distinguish the adult bees from the brood masses, Meikle et al. [36] proposed an approach based on continuous weight and temperature monitoring. To develop a predictive model, they systematically recorded data on hive weights and surface densities of frames with adult bees, capped/uncapped honey, and capped broods. Mass data obtained on adult bees and brood from hive inspections were regressed on the amplitudes of sine curves that fit the detrended temperature data. Weight data amplitudes were significantly correlated with adult bee populations (r^2^ = 0.82, *p* < 0.0001) during nectar flows, and temperature amplitudes were found to be inversely correlated with the log-transformed data of brood weight (r^2^ = 0.65, *p* < 0.01). Remote knowledge of daily honey production is crucial to estimate with a certain degree of accuracy the start and end of the nectar inflow and the amount of honey that could be retrieved from each hive/apiary, as well as to correlate any production failure to external factors (e.g., undesired weather events). In this respect, Catania and Vallone [35] monitored the main environmental factors, both inside and outside the hive, to assess their influence on the daily honey production. Their platform, founded on Arduino^®^ technology, measured internal and external air temperature, humidity, and the weight of the hive as well. In that trial, honey production (5 Kg in 18 days from the placement of the hives in the fields) was extrapolated by zeroing the hive scale once applied (this approach, however, does not exclude the problem of taking into account the mass of expansion of the colony). The data set acquired in that work allowed for valid decision support for the operator, such as the time of addition of supers or their removal at the end of the import of nectar. To date, there are no PB systems that allow us to estimate exclusively the supers’ weight, and therefore there is no way to know directly and accurately the production of honey; this could be a crucial aspect in the development of new PB systems in the future. To gain an overall picture of the behavior of the hive, it can also be beneficial to acquire information about the flight activity of bees. Using object tracking and Machine Learning, Fruet et al. [88] were able to count and monitor ongoing and outgoing worker bees and drones with an average error of 5% compared to the manual count made by operators on 27 videos captured from the entrance of 24 beehives. This prototype, called *ApisFlow*, represents a non-invasive, fairly accurate PB system for the honeybees’ flight activity.

Danieli et al. [89] tried to assess the ongoing and outgoing flight count accuracy of a prototypal device based on infrared cells’ technology by using a manual bee simulation and by manual counting of bees’ flight recorded through a camera trap. Both methods allowed them to assess that the flight sensor was unreliable, being affected by a high asymmetrical distribution between incoming and outgoing flights that made it impossible to find a simple data-correction model to be implemented at the hive or server level. Fiedler et al. [90], and subsequently Wakjira et al. [91], tried to apply the same PB systems prototype, called SAMS HIVE, to try to find valid solutions to some Ethiopian and Indonesian management problems, mainly due to the great distance between the hives, often located in deep forests, or the aggressive behavior of local honeybees that make remote monitoring of the hives themselves useful. The initial PB system design [90] featured a horizontal frame incorporating internal and external sensors for monitoring temperature and humidity, positioned beneath the first super, alongside a scale for tracking the total hive weight. Subsequently, this design was enhanced by integrating the sensor-equipped frame directly into the brood chamber [91]. The SAMS HIVE allowed for a decrease in the number of on-site observations of the bee colonies by beekeepers concerning unmonitored hives, reducing the risk for the operator, who can only carry out inspections if necessary. It helps to detect abnormal behaviors of the colonies and could help to prevent death, absconding, and cases of colony collapse disorder (CCD) as well. Implementation of the bee colony monitoring system to support the development of PB in remote areas can have added value for the beekeepers by increasing their honey production and subsistence. To predict swarming events, the monitoring of hive sounds could be a useful tool. As an example, Ferrari et al. [41] analyzed sound characteristics in beehives and their correlation with temperature and humidity changes by studying 270 h of sound recordings in three beehives alongside temperature and humidity data. During increased bee activity, like swarming, sound frequencies changed from about 100–300 Hz to 500–600 Hz, attributed to the bees’ vigorous wing flapping, resulting in a temperature drop from 35 °C to 33 °C [41]. In a recent study, Danieli et al. [92] remotely identified three distinct swarming events, validated through as many swarms captures in the field, by monitoring weight loss (1.1–2.1 Kg) using a commercial PB system (Melixa S.r.l., Trento, Italy). As found by others [93], a hive temperature increase (1–2 °C) was detected in the first part of each swarming event. Combining in-hive temperature increases and hive weight decreases can improve the overall reliability of swarm management practices based on the PB system’s warnings.

Optimization of technical problems is a focal point in the development of the systems to be adopted in PB. Energy consumption of data acquisition systems, with particular consideration given to installations in remote locations lacking a readily available energy supply, has a pivotal role in the optimization and reliability of PB systems. In this context, Hadjur et al. [68] measured the consumption and thermal performance (working CPU’s temperature during the data acquisition and transmission) of a Raspberry Pi^TM^ chip within a PB system in different environmental conditions (room temperature, hot and cold field test). According to that study, it seems that the working temperature of the system does not significantly affect the energy consumption of the execution of a script. In addition, the current approach to acquiring data, which consists of completely shutting down the system and waking it every hour to perform data collection, seems to be an efficient energy solution because the residual consumption of an inactive Raspberry Pi device has a significant impact on energy consumption [68].

The accuracy of data measurements, especially in relation to the weight and temperature of the beehive, can be critical for the beekeeper in the surveillance of colony well-being and honey production. Danieli et al. [94] showed that some simple data manipulation that can be easily implemented at the data generation (hive/apiary) or storage (cloud) level can enhance the accuracy of the informative potential of two commercial PB systems that, probably due to different sensors’ technology, exhibited slight but not-negligible differences. One system exhibited a relative deviation of 35% for weight increases (e.g., nectar flow) below 1 kg, but the other one demonstrated relatively substantial deviations of 25% for weights extending up to 6.5 kg [94]. The author addressed this issue by employing a straightforward approach to correlating the recorded data with known weights, thus facilitating the application of corrections to attain precise and highly usable data.

### 4.2. Market Options

Precision beekeeping systems are already commercially available products, and it is feasible to access producers’ websites via an online survey to obtain relevant information on offered options. This information can be beneficial not only for those involved in applied research but also, most importantly, for beekeepers. However, market options for such systems often lack comprehensive packages encompassing all the possible sensors with which a PB system could be equipped (Appendix A). In addition, the pricing is very often not available to a broad range of users, but it is expected to vary considerably for each system depending on how many sensors, which can usually be purchased separately, the beekeeper is interested in. The most common system (equipped with scale, temperature, and humidity sensors) has an average cost of 422 EUR/hive (216–874 EUR/hive), calculated on 16 of 32 PB systems found on the web market. A bit less than 70% of the PB system found were European, about a fifth were American, and a tenth were Asian, mostly from the Middle East and India, while the least represented were Australians (Appendix A). No data concerning Chinese or African PB systems can be easily retrieved due to the linguistic difficulty. Unfortunately, probably due to patenting, it was difficult to find detailed information about the types of sensors and technologies used in marketed PB systems. However, in about one case out of ten (90.6%), the sensors most integrated into PB systems were those measuring internal temperature, followed by weight (84.4%) and humidity (81.3%) (Figure 1). Overall, these sensors are among the earliest to have been developed and thus feature prominently in nearly all systems. A significant proportion of PB systems also include sound sensors (46.9%), which are frequently combined with internal temperature, weight, and humidity sensors. This four-sensor combination appears to be the most favored setup for PB systems worldwide. A low percentage (15.6%) of the systems incorporate flight activity recorders. Interestingly, among the PB systems considered (Appendix A), only one had the capability to provide data separately for the bees entering and leaving the hive. This distinction is noteworthy as, when complemented with weight sensors, it can aid in detecting phenomena such as robbing, swarming, or external perturbative factors, such as the ones that can affect the orientation capability of the foragers [95]. Furthermore, less commonly found sensors in PB systems include cameras (3.1%) to monitor flight activity or theft, and light sensors (3.13%) as theft or accident indicators if the hive is open. The need for remote beehive monitoring and data collection is a crucial resource in scientific research. In Europe, the most (and only) widespread PB system used in published scientific research is the one developed and marketed by Melixa S.r.l. (Trento, Italy). This system is distinguished by its ideal alignment with the requirements of the research community, offering a basic suite of sensors such as temperature, humidity, weight, flight activity, and weather station that collectively facilitate a comprehensive analysis of the dynamics of the hive in the context of each research experiment. Fontana et al. [96] use this PB system to relate flight activity and weight with the health status of colonies exposed to crop-protection pesticides in relation to the presence or absence of contaminants in the main matrices of the hive (wax, pollen, and honey). More recently, the application of the sensor suite of the Melixa PB system (Melixa s.r.l., Trento, Italy) has allowed Bota et al. [97] to find out that the flight activity of the colonies was negatively related to the vocal activity of the bee-eaters (*Merops apiaster* L.), detected by a passive acoustic monitoring system outside the apiary, providing insights about the possible role of PB systems in assessing damages caused by wildlife on apiaries. The same PB system was used by Danieli et al. [92], who detected some swarming events using hive weight decreases combined with in-hive temperature. It is worth noting that none of the marketed PB systems considered here is equipped with real-time, in-app alarm systems for events like swarming (both prevention and response) or robbing. Incorporating such systems could streamline the work of beekeepers significantly.

## 5. SWOT Analysis on the Application of PB Systems

The pros and cons of using PB systems can be systematically assessed according to a four-box analysis framework approach: strengths, weaknesses, opportunities, and threats (SWOT analysis) [98], as already performed by others [99,100,101]. As far as the adoption of PB systems in modern apiculture and in apidology research as well, the main factors that can be considered in a SWOT analysis are summarized in Table 1.

### 5.1. Strengths

The use of advanced sensors and Machine Learning algorithms [41,87,88] provides a wealth of accurate information regarding life inside the beehive [102]. Sensors’ data fusion enables timely hive intervention, reducing response time to issues like stress, diseases, and pests and enhancing beekeeping efficiency [18,103]. For example, this means preventing, with a combination of hive temperature or sound data, a possible swarming or being alerted in time, using collected scale data, that swarming has occurred [35,41,51,104,105]. At the same time, many systems are equipped with GPS and alarm systems, which warn the beekeeper in case the hive is stolen [106] and/or subject to accidents such as overturning due to wind, agricultural machining, or wildlife [90,97,107]. In practical applications, PB systems can support the beekeeper by optimizing honey harvesting [108] and by allowing product losses to be minimized [35,109].

### 5.2. Weaknesses

Precision beekeeping systems’ pricing is one of the main critical points that hinders the large-scale use of this type of equipment in modern apiculture [22]. These systems have a high cost, with components and sensors to be purchased separately, and they can raise the price of a complete system significantly. In addition to the purchase price, additional costs have also to be added for systems’ maintenance, troubleshooting by technical assistance services, and related shipment costs, which are difficult to calculate a priori [19]. In addition, it is important to note the difficulty in mass producing these systems on a global scale due to the current low customer demand, differences in hive construction, and the differences in honeybee species (e.g., *A. mellifera* vs. *A. cerana*) and/or subspecies and their behaviors [19,22,110]. It should also be noted that beekeepers hardly find information regarding the return on investment (ROI) of commercial PB systems [91]. In the survey conducted on products currently available on the market (see Section 4.2 of this review), only one (BeeSage B.V., Den Haag, The Netherlands) out of thirty-two PB systems found on the web market can give the beekeeper information regarding the ROI, which is estimated in two years for an apiary consisting of 10 hives. The energy autonomy and data connection of these devices are very relevant issues as well [22]. In fact, for the optimal use of the PB devices by beekeepers, it is essential that the system can power itself even in different climatic conditions [27,111] and can provide continuous real-time data to the owner. Though many of the commercial PB systems are equipped with photovoltaic panels, the climate and/or hive location (e.g., near trees, walls, buildings, etc.) cannot always allow for optimal charging of the batteries. The challenge of batteries, compounded by the potential for weak signals, poses a greater concern for nomadic beekeepers. A remote monitoring system for their hives would be highly advantageous. However, these beekeepers frequently relocate their hives to remote or wooded areas, where both power supply limitations and signal strength can hinder data transmission [25,112,113]. Another critical issue is the beekeeper’s preparedness for the proficient use of these technological tools. In fact, beekeepers without basic/advanced training in the use of hi-tech equipment may not always be able to read and evaluate data correctly or take advantage of using software for data manipulation. This issue could affect the major benefits these systems can give to the beekeeping of the future, data understanding, and improved production [19]. In some way, this issue can hamper the wide diffusion of these systems, by which a decrease in selling price can be reasonably expected. Finally, the accuracy of some systems, such as in- and out-flight traffic and sound detection [89,114], remains to be greatly improved before they can be effectively deployed for management purposes.

### 5.3. Opportunities

Hive sensors generate valuable continuous data that is impossible to obtain otherwise, offering a unique opportunity for research on honeybee colony behavior and their environment [17,19,36]. From a sustainability standpoint, a careful reading of the data sent to the beekeeper can be of help for shrewd hive management (according to the precision beekeeping paradigm itself), allowing fewer on-field beekeepers’ interventions, reducing unnecessary hive monitoring trips, and more judicious use of treatments as well [18,108]. The combination of WSN, modern technologies of the Internet of Things (IoT), and a user (in general, the beekeeper) can represent the most suitable way to support or substitute the manual visits to the apiary. The mass diffusion of these systems could also lead to further technological innovations in the beekeeping field. At the same time, a greater diffusion of this type of product could likely lead to a reduction in prices, boosting a greater diffusion of them, as has happened in the economies of scale of other technological products in recent decades [115].

### 5.4. Threats

To date, not all the available PB systems on the market offer the accuracy needed to correctly manage apiaries remotely. In principle, it could be a mistake to rely entirely on technology while neglecting the direct field experience that beekeepers have [114], as, all in all, they still play a crucial role in the proper management of an apiary. Considering also that, due to the high cost, it is economically unfeasible for a beekeeper to place more than one/two systems per apiary (overall, less than 10% of hives could be effectively monitored), reducing thus the real value of PB systems that can be well capitalized if all, at least, or a great part of the beehives of an apiary are equipped with them. As a possible result, the beekeepers can obtain an unreal perception of the more general conditions (production, colonies’ health status, etc.) of their apiaries. It also remains to be thoroughly evaluated to determine what impact these systems have on beehive life. Some studies have shown that hives exposed to low-frequency electromagnetic fields may be more affected by stress symptoms [116,117,118,119]. In Henry et al. [76], a group of three hives was subjected to a 2.4 GHz Wi-Fi signal (the most common Wi-Fi signal) for 30 days. In that experience, in comparison to a control group of three hives, an increase in temperature and humidity inside the hives was observed during the trial period, even though such differences were not statistically supported. In Odemer and Odemer [120], queen bee larvae in two hives were exposed to radiation from a common GSM device (which is often used by precision beekeeping systems to send data) during all life stages, evaluating hatching and mating success. The hatching ratio was reduced in the radiation-affected colonies, but no interference in mating was shown. However, few studies [120,121,122,123,124] have set out to assess the impacts of these waves and the electromagnetic radiation on beehives, although the topic needs in-depth study.

## 6. Mid-Term Perspectives and Conclusions

Indubitably, the development of PB systems will move traditional beekeeping into the mainstream of the Precision Livestock Farming (PLV) paradigm, moving from operative farmers’ choices based almost entirely on their own experience to decisions driven by quantitative data [125]. In this respect, the aims and goals of PB are similar to those that animated the development of PLF in other livestock sectors, such as, for example, real-time measuring of milk production in dairy herds [126] or body weight increase in broiler husbandry [127,128] to be relevant tools by which to plan and implement precision management strategies to improve productivity and maintain high animal welfare standards as well [129]. However, beekeeping can be regarded as a particular husbandry activity mainly due to the peculiarity of the farmed animal: the honeybee, a not fully domesticated species [130]. In some respects, beekeeping and sheep farming share similar pasture-based management with more limitations for the beekeeper than the sheep farmer in supporting the animals under inclement weather conditions or other unfavorable situations (e.g., seasonality). On the other hand, pasture-based animal production systems show several constraints that limit adopting PLF solutions [129], as the beekeeping shows: the distances from the farmer center, the uneven access to the communication web, and, very often, the unavailability of electricity supply grids to power electronic devices. Roughly in line with the potentiality of the commercial solutions available to date, the state of the art described in this review allows for an overview of the first key point of PB [17]: data collection. However, even from that point of view, more R&D is required and certainly is expected to take place in the next few years on the real-time monitoring of some relevant hive activities and behavioral features such as in–out flight trafficking, in-hive air composition (i.e., CO_2_ concentration), or colony sound/vibrations. Looking at the real development and implementation of PB, as for other measurable quantities of a hive (weight, humidity, temperature), gathering data is of utmost importance but, per se, has few if any practical utility if not adequately supported by data processing, but also by data fusion as well, that can effectively support beekeepers in gaining integrated insights more easily and more deeply than the ones they routinely obtain by visiting apiaries and hives. In this respect, further improvements will be deeply rooted in the experiential background of skilled beekeepers (mainly the professional ones) and, wherever possible, in the experience gained in other fields of PLF. In some circumstances, due to time or weather conditions, the skilled beekeeper can obtain a general overview of the status of a colony by knocking on the side of the hive and hearing the bees’ sound response. In this respect, upgrades of commercial PB systems should encompass the possibility for the beekeeper to check at a distance the status of his/her hives by stimulating them through any type of knocking-type cue and evaluating the related outcomes (i.e., sound/vibration reply) directly or mediate through any sort of decision support system [25]. At the same time, Machine Learning is widely studied in PLF applications for cattle [125] or poultry production [131], but to date, there are few reports in the PB-related literature [84,85] and very few, if any, commercial solutions. To be a very practical and effective option for beekeepers, PB has to aim for tangible improvements not only in apiary productivity, a very relevant issue, especially for practitioner beekeepers, but also in beehive welfare, which includes indubitably the proper management of honeybee pests and parasites [132]. From this specific standpoint, the main R&D pipelines should address some relevant scientific questions and practical issues such as the relationships among food source availability, bee food quality, climate conditions, other environmental stressors, bees’ genetics, and the outbreak of hive pathologies (i.e., varroasis, nosemiasis, and virosis), all of which are topics of outstanding relevance to safeguarding honeybees as pollinating insects, for crops and wild flora as well, but also as farm animals.

## Figures and Tables

**Figure 1 animals-14-00070-f001:**
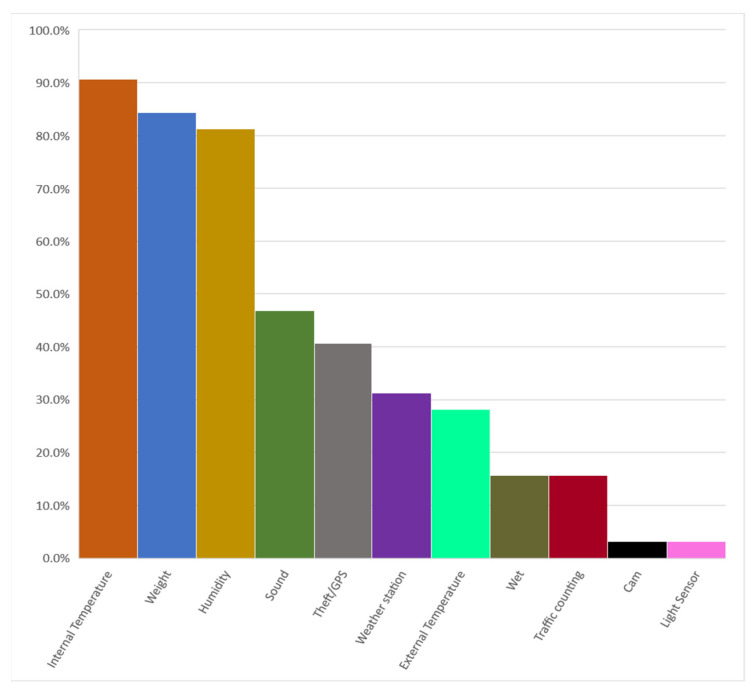
Types of sensors fitted on commercial precision beekeeping (PB) systems available on the web market.

**Table 1 animals-14-00070-t001:** SWOT analysis with Internal (Strengths, Weaknesses) and External (Opportunities, Threats) factors on the use of precision beekeeping (PB) systems in beekeeping.

Strengths	Weaknesses
Accurate information Preventing and monitoring troubles and swarming Improving productivity	Cost Energy efficiency Beekeepers’ preparedness Sensor accuracy
Opportunities	Threats
Management innovations	Technology dependence
Time-saving	Bees’ health threats
Bee knowledge improvement	

## Data Availability

All data are included in the article.

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
