# Peer review of "Precision Beekeeping Systems: State of the Art, Pros and Cons, and Their Application as Tools for Advancing the Beekeeping Sector"

_animals, 2023, doi:10.3390/ani14010070_

Round 1
Reviewer 1 Report
Comments and Suggestions for Authors
The review carried out by the authors is extremely interesting, the topic is considered taking into account relevant information and suggesting the points that should be improved in the search for greater production of beekeeping systems. Some quotes that are missing on lines 203-206 and lines 238-240.
Author Response
Dear Reviewer, responses to your comments are in the "review report" highlighted in red, while changes to the "reviewed manuscript" are highlighted in yellow. Please see the attachment.
Reviewer 1
The analysis carried out by the authors is extremely interesting,
the topic is addressed taking into account the relevant information
and suggesting points that should be improved in the pursuit of greater production of beekeeping systems.
Au: Dear reviewer, thank you very much for your comment.
In lines 203-206 and 238-240 some citations are missing.
Au: Dear reviewer, thank you for your suggestion, we added it in the text.

Reviewer 2 Report
Comments and Suggestions for Authors
This review provides an interesting overview on the different smart technologies to monitor hives for productive and scientific purposes. The topic is very meticulously and well handled, but I would re-arrange and explain better some parts. I would moreover improve English language style and double check grammar and phrasing. For this reason, I would suggest accepting the work after a revision.
Simple Summary
Line 12-13 “ON” the results. Anyway lines 12-15 result redundant. Maybe just say: “This review aims to raise the attention on precision technologies applied…”
Abstract
The abstract is quite long, and some parts could be cut. For example, it is true that pollinators provide pollination, but I would avoid mentioning it, since it is also not the topic of the review. In general, I would try to improve the phrasing, reducing the length. For example, you could start the abstract directly from line 20 (“Honey bee husbandry….). Also, the aim of the review should be clearly written before starting to explain the results. Is the aim mentioned at line 34-37? Does “also” means there is another aim? In general, the abstract should be structured a bit better, with the aim of the review work more clearly stated.
Keywords
Some keywords are redundant.
1. Introduction
Phrase 61-63 should be better structured and shorten.
Line 66-70: in my opinion the conclusion is a bit too “drastic”… increasing the honeybees does not mean the resolution of the problem of biodiversity. The topic is very delicate, and many things should be considered (competition domestic-wild pollinators; diffusion of pathogens from domestic to wild etc. ) I would avoid handling this topic, that would be a review its-self. I would just say that beekeeping is becoming important in the framework of the decline of pollinators and that is going through changes due to new technologies.
Line 71-78: for me also this part is a bit “out of topic” and hastily treated. The introduction could start directly from line 79.
Line 110-114: I would better expose the aims of the review. Was it also done a SWOT analysis; wasn’t it? I would mention it here then. Please explain better the means and the goals of your work.
2. Data collection and storage: sensors and systems
This paragraph is interesting but a bit difficult to follow. Line 129-136 should be better framed in the paragraph. Is it an example of an “old” hive tech?
2.1 Weight
Part 185-188 is not clear for me. Also, could you explain better the definition “relatively limited global prevalence” and “higher ubiquity” at line 182-183?
2.2 Temperature
Mention please also thermo-regulation at line 205, that is connected to the health of the hive.
2.3 sounds
add please the work by Ferrari, S., Silva, M., Guarino, M., & Berckmans, D. (2008). Monitoring of swarming sounds in bee hives for early detection of the swarming period. Computers and electronics in agriculture, 64(1), 72-77.
4. Study cases and market options
This part is well done, but I would avoid the sub-paragraphs or re-arrange them a bit. For example, some part of honey bees’ behaviour and colony status could be together, while the monitoring of remote apiaries could go alone, or they could become a single paragraph.
5. SWOT analysis on the application of PB systems
In the SWOT analysis I would insist more on the “connectiveness” weakness, at least mentioning that it could be a problem for beekeepers practicing nomadism, in remote areas as mountains, that are indeed the ones needing the most a substitutive monitoring system compared to the direct inspection of the hives.
References
Please check all the recent literature is cited. For example, it seems to me that “Anwar, O., Keating, A., Cardell-Oliver, R., Datta, A., & Putrino, G. (2022). Design and development of low-power, long-range data acquisition system for beehives-BeeDAS. Computers and Electronics in Agriculture, 201, 107281” is missing.
This review provides an interesting overview on the different smart technologies to monitor hives for productive and scientific purposes. The topic is very meticulously and well handled, but I would re-arrange and explain better some parts. I would moreover improve English language style and double check grammar and phrasing. For this reason, I would suggest accepting the work after a revision.
Simple Summary
Line 12-13 “ON” the results. Anyway lines 12-15 result redundant. Maybe just say: “This review aims to raise the attention on precision technologies applied…”
Abstract
The abstract is quite long, and some parts could be cut. For example, it is true that pollinators provide pollination, but I would avoid mentioning it, since it is also not the topic of the review. In general, I would try to improve the phrasing, reducing the length. For example, you could start the abstract directly from line 20 (“Honey bee husbandry….). Also, the aim of the review should be clearly written before starting to explain the results. Is the aim mentioned at line 34-37? Does “also” means there is another aim? In general, the abstract should be structured a bit better, with the aim of the review work more clearly stated.
Keywords
Some keywords are redundant.
1. Introduction
Phrase 61-63 should be better structured and shorten.
Line 66-70: in my opinion the conclusion is a bit too “drastic”… increasing the honeybees does not mean the resolution of the problem of biodiversity. The topic is very delicate, and many things should be considered (competition domestic-wild pollinators; diffusion of pathogens from domestic to wild etc. ) I would avoid handling this topic, that would be a review its-self. I would just say that beekeeping is becoming important in the framework of the decline of pollinators and that is going through changes due to new technologies.
Line 71-78: for me also this part is a bit “out of topic” and hastily treated. The introduction could start directly from line 79.
Line 110-114: I would better expose the aims of the review. Was it also done a SWOT analysis; wasn’t it? I would mention it here then. Please explain better the means and the goals of your work.
2. Data collection and storage: sensors and systems
This paragraph is interesting but a bit difficult to follow. Line 129-136 should be better framed in the paragraph. Is it an example of an “old” hive tech?
2.1 Weight
Part 185-188 is not clear for me. Also, could you explain better the definition “relatively limited global prevalence” and “higher ubiquity” at line 182-183?
2.2 Temperature
Mention please also thermo-regulation at line 205, that is connected to the health of the hive.
2.3 sounds
add please the work by Ferrari, S., Silva, M., Guarino, M., & Berckmans, D. (2008). Monitoring of swarming sounds in bee hives for early detection of the swarming period. Computers and electronics in agriculture, 64(1), 72-77.
4. Study cases and market options
This part is well done, but I would avoid the sub-paragraphs or re-arrange them a bit. For example, some part of honey bees’ behaviour and colony status could be together, while the monitoring of remote apiaries could go alone, or they could become a single paragraph.
5. SWOT analysis on the application of PB systems
In the SWOT analysis I would insist more on the “connectiveness” weakness, at least mentioning that it could be a problem for beekeepers practicing nomadism, in remote areas as mountains, that are indeed the ones needing the most a substitutive monitoring system compared to the direct inspection of the hives.
References
Please check all the recent literature is cited. For example, it seems to me that “Anwar, O., Keating, A., Cardell-Oliver, R., Datta, A., & Putrino, G. (2022). Design and development of low-power, long-range data acquisition system for beehives-BeeDAS. Computers and Electronics in Agriculture, 201, 107281” is missing.
Comments on the Quality of English Language. I would improve English language style and double check grammar and phrasing.
Author Response
Dear Reviewer, in the review report the responses to your comments are reported here highlighted in red, while the revised manuscript is attached in which the changes made are highlighted in yellow. Please see the attachment.
Reviewer 2
This review provides an interesting overview on the different smart technologies to monitor hives for productive and scientific purposes. The topic is very meticulously and well handled, but I would re-arrange and explain better some parts.
Au: thank you very much for your comment.
I would moreover improve English language style and double check grammar and phrasing. For this reason, I would suggest accepting the work after a revision.
Au: the English language has been checked.
Simple Summary
Line 12-13 “ON” the results. Anyway lines 12-15 result redundant. Maybe just say: “This review aims to raise the attention on precision technologies applied…”
Au: the simple summary has been re-arranged according to your suggestion.
Abstract
The abstract is quite long, and some parts could be cut. For example, it is true that pollinators provide pollination, but I would avoid mentioning it, since it is also not the topic of the review. In general, I would try to improve the phrasing, reducing the length. For example, you could start the abstract directly from line 20 (“Honey bee husbandry….). Also, the aim of the review should be clearly written before starting to explain the results. Is the aim mentioned at line 34-37? Does “also” means there is another aim? In general, the abstract should be structured a bit better, with the aim of the review work more clearly stated.
Au: the abstract has been shorted and re-organized. The aims are reported at the beginning of this section.
Keywords
Some keywords are redundant.
Au: key words have been partly changed to avoid repletion of words in the title.
- Introduction
Phrase 61-63 should be better structured and shorten.
Au: the period has been modified and shortened.
Line 66-70: in my opinion the conclusion is a bit too “drastic” … increasing the honeybees does not mean the resolution of the problem of biodiversity. The topic is very delicate, and many things should be considered (competition domestic-wild pollinators, diffusion of pathogens from domestic to wild etc.) I would avoid handling this topic, that would be a review its-self. I would just say that beekeeping is becoming important in the framework of the decline of pollinators and that is going through changes due to new technologies.
Au: you are right. The period has been modified and now is less “drastic”.
Line 71-78: for me also this part is a bit “out of topic” and hastily treated. The introduction could start directly from line 79.
Au: We don't completely agree with your consideration. The problems that honey bees encounter represent one of the reasons why constant and real-time monitoring can be useful, and it is right to mention them. You're right though that we've gone a little too long on this topic. We proceeded to reduce it trying to better clarify our point of view.
Line 110-114: I would better expose the aims of the review. Was it also done a SWOT analysis; wasn’t it? I would mention it here then. Please explain better the means and the goals of your work.
Au: The aims of our review have been better clarified.
- Data collection and storage: sensors and systems
This paragraph is interesting but a bit difficult to follow. Line 129-136 should be better framed in the paragraph. Is it an example of an “old” hive tech?
Au: thank you for your comment, now all is well explained (we hope!).
2.1 Weight
Part 185-188 is not clear for me. Also, could you explain better the definition “relatively limited global prevalence” and “higher ubiquity” at line 182-183?
Au: The period has been rewritten (lines 177-180).
Au: “higher ubiquity” and “relatively limited global prevalence” have been deleted.
2.2 Temperature
Mention please also thermo-regulation at line 205, that is connected to the health of the hive.
Au: added (line 197).
2.3 sounds
add please the work by Ferrari, S., Silva, M., Guarino, M., & Berckmans, D. (2008). Monitoring of swarming sounds in bee hives for early detection of the swarming period. Computers and electronics in agriculture, 64(1), 72-77.
Au: added. See lines 236-238 (Ref. [41]).
- Study cases and market options
This part is well done, but I would avoid the sub-paragraphs or re-arrange them a bit. For example, some part of honey bees’ behaviour and colony status could be together, while the monitoring of remote apiaries could go alone, or they could become a single paragraph.
Au: thanks for your comment, the part has been reorganized into only two paragraphs: "Case Studies" and “Market options". We think the division between the two topics needed to better explain the origin of the commercially available PB systems starting from the development of prototypes.
- SWOT analysis on the application of PB systems
In the SWOT analysis I would insist more on the “connectiveness” weakness, at least mentioning that it could be a problem for beekeepers practicing nomadism, in remote areas as mountains, that are indeed the ones needing the most a substitutive monitoring system compared to the direct inspection of the hives.
Au: Thank you for the advice. The aspect of nomadism and the potential support that precision apiculture systems could provide to this sector is very intriguing, although it is not extensively studied. A small section has been added to address this in the paragraph discussing weaknesses.
References
Please check all the recent literature is cited. For example, it seems to me that “Anwar, O., Keating, A., Cardell-Oliver, R., Datta, A., & Putrino, G. (2022). Design and development of low-power, long-range data acquisition system for beehives-BeeDAS. Computers and Electronics in Agriculture, 201, 107281” is missing.
Au: Thank you for the suggestion. The quote has been added (Ref. 111).

Round 2
Reviewer 2 Report
Comments and Suggestions for Authors
Revisions request was sufficiently satisfied.